# An Adaptive and Robust Control Strategy for Real-Time Hybrid Simulation

**DOI:** 10.3390/s22176569

**Published:** 2022-08-31

**Authors:** Hong-Wei Li, Fang Wang, Yi-Qing Ni, You-Wu Wang, Zhao-Dong Xu

**Affiliations:** 1Department of Civil and Environmental Engineering, The Hong Kong Polytechnic University, Hung Hom, Kowloon, Hong Kong, China; 2Key Laboratory of C&PC Structures of the Ministry of Education, Southeast University, Nanjing 211189, China; 3School of Civil Engineering, China-Pakistan Belt and Road Joint Laboratory on Smart Disaster Prevention of Major Infrastructures, Southeast University, Nanjing 211189, China

**Keywords:** sliding mode control, bounded-gain forgetting, least-squares estimator, robustness, adaptation, real-time hybrid simulation, benchmark

## Abstract

A real-time hybrid simulation (RTHS) is a promising technique to investigate a complicated or large-scale structure by dividing it into numerical and physical substructures and conducting cyber-physical tests on it. The control system design of an RTHS is a challenging topic due to the additional feedback between the physical and numerical substructures, and the complexity of the physical control plant. This paper proposes a novel RTHS control strategy by combining the theories of adaptive control and robust control, where a reformed plant which is highly simplified compared to the physical plant can be used to design the control system without compromising the control performance. The adaptation and robustness features of the control system are realized by the bounded-gain forgetting least-squares estimator and the sliding mode controller, respectively. The control strategy is validated by investigating an RTHS benchmark problem of a nonlinear three-story steel frame The proposed control strategy could simplify the control system design and does not require a precise physical plant; thus, it is an efficient and practical option for an RTHS.

## 1. Introduction

Full-scale structural tests are inefficient and expensive, and they are limited by the capacity of the testing equipment (actuator or shake table). A real-time hybrid simulation (RTHS) provides a new option to conduct structural tests, where the large part of the structure that is well understood is extracted as the *numerical substructure* and modeled on a computer, while the rest of structure that is normally a small part with complicated dynamic properties is treated as the *physical substructure* and manufactured for testing [1,2,3]. An RTHS is much more efficient than full-scale tests and has been widely used for a structural analysis [4,5,6,7,8].

A loading device, such as a servo-hydraulic actuator or a shake table, works as the *transfer system* to simulate the boundary conditions between the numerical and physical substructures. The key of an RTHS is to design a controller/control system to drive the physical plant which includes the transfer system and physical substructure to track designated motions, where time delays/lags, sensor noises, modeling uncertainties and disturbances should be well handled [9]. Compared to a normal control problem where the designated motions are not influenced by the control performance, the control system design for an RTHS is more challenging because there is an additional feedback loop between the physical and numerical substructures that could accumulate errors in the designated motions, and the physical plant is normally very complicated. In the recent decades, the research community has paid more attention to a model-based control design that could compensate the time delays/lags and reject noises and uncertainties [10,11,12,13]. Nevertheless, to the best of the authors’ knowledge, most of the past studies assumed that the model of the physical plant can be obtained, and even if uncertainties were considered, the model parameters were varying in known boundaries. We believe that such information might not be available in practice because the motivation of the RTHS is to conduct dynamic tests on the part of a structure that is not well understood. More focus should be placed on developing RTHS control systems for the situation where the knowledge of the physical plant is very limited.

In this paper, a control strategy taking advantage of both adaptive control and robust control is proposed, which allows us to use a reformed plant instead of the physical plant to design the RTHS control system. The adaption and robustness properties are realized by the bounded-gain forgetting least-squares (BLS) estimator and the sliding mode controller, respectively. At first, the methodologies of the BLS estimator and the sliding mode controller are presented, and the control strategy is demonstrated by a nonlinear illustrative example. Next, the proposed control strategy is applied to a well-defined virtual RTHS benchmark problem [14], where a small portion of the first floor of a three-story steel-frame structure is extracted as the physical substructure, and the remaining portion of the structure is treated as the numerical substructure. Different physical plants are created by adding a damper or creating damage in the physical substructure. A single control system is designed for these conditions, based on a unique reformed plant. The simulation results show that high RTHS global tracking performances are achieved for all the three reference structures, which validates the effectiveness of the proposed control strategy for the RTHS.

## 2. Adaptive and Robust Control Strategy

The *n*-th order uncertain linear or nonlinear control plant is
(1)$$x^{\left(n\right)}=-\alpha f\left(x,\dot{x},\cdots ,x^{\left(n-1\right)}\right)+bu$$
where
(2)$$\alpha =\left[\alpha_{1},\alpha_{2},\cdots ,\alpha_{m}\right],\;f\left(x,\dot{x},\cdots ,x^{\left(n-1\right)}\right)={\left[f_{1},f_{2},\cdots ,f_{m}\right]}^{T},$$
*u* is the command input, α and *b* are unknown parameters (constant or slowly time varying), $f_{1}$, $f_{2}$, ⋯, $f_{m}$ are linear or nonlinear functions where the derivative orders are equal to or less than n−1, the state variable $x^{\left(n-1\right)}$ which has the superscript (n−1) means the (n−1)-th-order derivative of the first state *x*. The control target is to design a control system to generate the input command *u* that can drive the system’s state vector $x={\left[x,\dot{x},\cdots ,x^{\left(n-1\right)}\right]}^{T}$ to follow a designated trajectory $x_{d}={\left[x_{d},{\dot{x}}_{d},\cdots ,x_{d}^{\left(n-1\right)}\right]}^{T}$, regardless of model uncertainties and measurement noises.

The control strategy proposed in this paper contains two layers: an adaptation layer and a robustness layer. In the adaptation layer, unknown parameters α and *b* are estimated online based on the plant’s past input and output data, by using the BLS estimator. The estimations are denoted as $\hat{\alpha }$ and $\hat{b}$. Then, in the robustness layer, a sliding mode controller is designed assuming that there are bounded errors between the estimations and true values of the model parameters. It is important to remark that for the control plant that could be well modeled with bounded and slowly time-varying parameters, either of the adaptation and robustness layers would be sufficient to achieve a high control performance. However, the control strategy we proposed requires both layers to deal with more intractable control plants as we will discuss later in detail. Both the BLS estimator and the sliding mode controller have been proposed and discussed in our previous work [15,16,17]. Here, we will briefly present how the BLS estimator and the sliding mode controller work and then describe the control strategy that could fully take advantage of them. At the end of this section, an illustrative example is conducted for demonstration purpose.

### 2.1. Adaptation Layer

To estimate the model parameters, the control plant is rewritten as the following form:(3)y=WΨ
where y and W are calculated from the input signal *u* and measurements; Ψ contains all the model parameters in sequence or expressions of them. The most simple configuration is given by
(4)$$y=x^{\left(n\right)},\;W=\left[-f,u\right],\;\Psi ={\left[\alpha ,b\right]}^{T},$$
where the full states from *x* to $x^{\left(n\right)}$ need to be observable. In most cases, only limited states (normally low-order states) could be directly measured, e.g., *x*, and sensor noises are contained. Therefore, Equation (Equation 3) could be constructed by using some tricks, such as applying both sides of Equation (Equation 1) through a low-pass filter to get rid of high-order states and measurement noises. This process will be demonstrated in Section 3.3.

The *estimation*, *predicted output*, *estimation error* and *prediction error* are denoted as $\hat{\Psi }$, $\hat{y}$, $\tilde{\Psi }$ and $e_{1}$, respectively. In addition, the following expressions are satisfied:(5)$$\hat{y}=W\hat{\Psi },$$
(6)$$\tilde{\Psi }=\hat{\Psi }-\Psi ,$$
(7)$$e_{1}=\hat{y}-y=W\tilde{\Psi }.$$

The typical least-squares method for parameter estimations is to minimize
(8)$$J_{p0}\left(t\right)=\int_{0}^{t}\|e_{1}{\left(q\right)\|}^{2}dq$$
where the function ‖·‖ returns the $l^{2}$-norm of a vector or matrix. This method considers all the past prediction errors and thus has a strong capability to estimate constant parameters with respect to disturbances and measurement, which will be well smoothed out during the estimation process [18]. On the other hand, it shows poor performance in tracking time-varying parameters because past data are generated by past parameters. To overcome this deficiency, the following error criteria
(9)$$J_{p}\left(t\right)=\int_{0}^{t}\Gamma \left(q,t\right)\|e_{1}{\left(q\right)\|}^{2}dq$$
are adopted instead of Equation (Equation 8). In Equation (Equation 9),
(10)$$\Gamma \left(q,t\right)=e^{-\int_{q}^{t}\theta (r)dr}$$
is the *weighting coefficient* which puts more attention on the recent data and decreases the influence of the remote data; θ(r)>0 is the time-varying *forgetting factor* that needs to be tuned based on the estimation errors.

The optimum estimation $\hat{\Psi }\left(t\right)$ should satisfy:(11)$$\hat{\Psi }\left(t\right)=P\left(t\right)\int_{0}^{t}\Gamma \left(q,t\right)W^{T}\left(q\right)y\left(q\right)dq,$$
where *estimation gain* P(t) is introduced here, given by
(12)$$P\left(t\right)={\left[\int_{0}^{t}\Gamma \left(q,t\right)W^{T}\left(q\right)W\left(q\right)dq\right]}^{-1}.$$
The first-order derivatives of $\hat{\Psi }\left(t\right)$ and P(t) could be deduced as follows:(13)$$\dot{\hat{\Psi }}\left(t\right)=-P\left(t\right)W^{T}\left(t\right)e_{1}\left(t\right),$$
(14)$$\dot{P}\left(t\right)=\theta \left(t\right)P\left(t\right)-P\left(t\right)W^{T}\left(t\right)W\left(t\right)P\left(t\right).$$
The derivations of Equations (Equation 13) and (Equation 14) are based on the following formulas:(15)$$\dot{\Gamma }\left(0,t\right)=\frac{d}{dt}\left[e^{-\int_{0}^{t}\theta (r)dr}\right]=-\theta \left(t\right)\Gamma \left(0,t\right),$$
(16)$$\frac{d}{dt}\left[P\left(t\right)P{\left(t\right)}^{-1}\right]=\dot{P}\left(t\right)P{\left(t\right)}^{-1}+P\left(t\right)\dot{P}{\left(t\right)}^{-1}=0,$$
and if
(17)$$F\left(t\right)=\int_{0}^{t}\Gamma (q,t)\Delta (q)dq=\Gamma \left(0,t\right)\int_{0}^{t}\frac{\Delta (q)}{\Gamma (0,q)}dq,$$
then
(18)$$\dot{F}\left(t\right)=-\theta \left(t\right)F\left(t\right)+\Delta \left(t\right).$$

According to Equation (Equation 12), P(0)=+∞, making it impossible to initiate the estimator in simulation. Therefore, Equation (Equation 12) is modified as
(19)$$P\left(t\right)={\left[\Gamma \left(0,t\right)\;P{\left(0\right)}^{-1}+\int_{0}^{t}\Gamma \left(q,t\right)W^{T}\left(q\right)W\left(q\right)dq\right]}^{-1},$$
where P(0) is defined manually (should be a positive definite matrix with relatively high eigenvalues). Because Γ(0,t) monotonically decreases and
(20)$$\lim_{t\to +\infty }\Gamma (0,t)=0,$$
Equation (Equation 19) is approaching Equation (Equation 12) as time increases; thus, the error caused by P(0) is gradually eliminated. The first-order derivatives of $\hat{\Psi }\left(t\right)$ and P(t) remain the same as Equations (Equation 13) and (Equation 14) with the adoption of Equation (Equation 19).

The forgetting factor is tuned using
(21)$$\theta \left(t\right)=\theta_{0}\left(1-\frac{\left\|P(t)\right\|}{k_{0}}\right)$$
where $\theta_{0}>0$, $k_{0}\geq \left\|P\left(0\right)\right\|$. Equations (Equation 13), (Equation 14) and (Equation 21) together form the BLS estimator. Larger values of $\theta_{0}$ and $k_{0}$ indicate “faster” forgetting, which could increase the estimator’s capability in tracking time-varying parameters. Nevertheless, higher weights on the small range of recent data points would enlarge the negative effects caused by noise, disturbance, modeling errors and uncertainties, and thus create more oscillations in estimations. Therefore, the selections of $\theta_{0}$ and $k_{0}$ involve a trade-off between the capabilities of tracking time-varying parameters and rejecting noise and disturbance.

### 2.2. Robustness Layer

The estimated values of the plant’s parameters: $\hat{\alpha }$ and $\hat{b}$ are obtained from $\hat{\Psi }$ and then are treated as the nominal values of plant’s parameters. The errors between nominal and true values are assumed to be bounded:(22)$$|\alpha -\hat{\alpha }|\leq \bar{\alpha },\;\left|b-\hat{b}\right|\leq \bar{b},$$
where $\bar{\alpha }$ and $\bar{b}$ are *estimation error boundaries*. Additionally, the sign of *b* is known and constant, and the signs of *b* and $\hat{b}$ are identical during the estimation, i.e.,
(23)$$b\hat{b}>0,\;|b|>0,\;|\hat{b}|>0.$$
This assumption is reasonable because the input gain *b* is normally constant or varying within a small range.

Sliding mode controllers have been widely adopted for the tracking control of systems which have bounded model uncertainties like Equation (Equation 22) [19,20]. The methodologies of the sliding mode controller are presented below. The *compact error* combining tracking errors of full states is defined as
(24)$$E={\left(\frac{d}{dt}+\lambda \right)}^{n-1}e=e^{\left(n-1\right)}+\sum_{i=1}^{n-1}\frac{(n-1)!}{i!(n-1-i)!}\lambda^{i}e^{\left(i-1\right)},$$
where λ is a positive constant, $e=x-x_{d}$ is the first state tracking error. For instance, n=3, $E=\ddot{e}+2\lambda \dot{e}+\lambda^{2}e$. Equation (Equation 24) indicates that *e* could be regarded as scalar obtained by applying the low-pass filter 1/(s+λ) on *E* for n−1 times, where *s* is the Laplace variable. It can be proven [18] that if there exists a small positive value ξ that satisfies
(25)|E(t)|≤ξ,∀t>0,
then we have
(26)$$|e^{\left(i\right)}\left(t\right)|\leq \frac{2^{i}\lambda^{i}}{\lambda^{n-1}}\xi ,\;\forall t>0,\;i=0,1,...,n-1.$$
Thus, the tracking of the states *x*, $\dot{x}$, …, $x^{\left(n-1\right)}$ is equivalent to stabilizing the single scalar *E*. It can be obtained that
(27)$$\dot{E}=-\alpha \;f+bu-h,$$
where
(28)$$h=x_{d}^{\left(n\right)}-\sum_{i=1}^{n-1}\frac{(n-1)!}{i!(n-1-i)!}\lambda^{i}e^{\left(i\right)}.$$

E=0 is regarded as the *sliding surface*, which is the perfect condition where there are no tracking errors. A sliding mode controller is trying to ensure that the distance from *E* to the sliding surface always tends to decrease, i.e.,
(29)$$\begin{array}{cc} \; & \dot{E}\leq 0\;\mathrm{for}\;E\geq 0,\\ \; & \dot{E}>0\;\mathrm{for}\;E<0, \end{array}$$
despite the presence of model uncertainties. Equation (Equation 29) can be reformed as the following inequality:(30)$$\dot{E}E\leq -\eta \left|E\right|$$
where η is a positive constant. Equation (Equation 30) can be also formed based on Barbalat’s lemma [18], by choosing an energy-type Lyapunov function
(31)$$V=\frac{1}{2}E^{2}$$
and making
(32)$$\dot{V}\leq -\eta \left|E\right|<0$$
to guarantee the boundedness of *E*.

The input command is designed as
(33)$$u=\hat{u}+u_{c},$$
where
(34)$$\hat{u}={\hat{b}}^{-1}\left(\hat{\alpha }\;f+h\right)$$
is the *nominal command* to achieve $\dot{E}=0$, assuming that the nominal model parameters $\hat{\alpha }$ and $\hat{b}$ are used in Equation (Equation 27), and $u_{c}$ is the *correction input* that is aimed to reject the divergences caused by parameter uncertainties. The ideal form of $u_{c}$ is
(35)$$u_{c}=-{\hat{b}}^{-1}k\;\mathrm{sgn}\left(E\right)$$
where *k* is the *correction gain* that needs to be tuned, and
(36)$$\mathrm{sgn}\left(E\right)=\left\{\begin{array}{cc} -1,\; & E<0,\\ 0,\; & E=0,\\ 1,\; & E>0. \end{array}\right.$$
However, the adoption of Equations (Equation 30) and (Equation 35) would create undesirable chatterings/oscillations around the sliding surface, which might excite unmodeled high-frequency dynamics of the control plant or even make the system unstable [15,18]. To avoid such a scenario, the boundary layer Φ>0 is used to smooth out the command input. The compact error *E* is driven back into the boundary layer Φ rather than the strict sliding surface E=0; therefore, the following relationships are established
(37)$$\begin{array}{cc} \; & \dot{E}\leq \dot{\Phi }\;\mathrm{for}\;E\geq \Phi ,\\ \; & \dot{E}>-\dot{\Phi }\;\mathrm{for}\;E<-\Phi , \end{array}$$
which can be rewritten as
(38)$$\dot{E}E\leq \left(\dot{\Phi }-\eta \right)\left|E|\;\mathrm{for}\;|E\right|\geq \Phi .$$
The correction input is modified as
(39)$$u_{c}=-{\hat{b}}^{-1}k\;\mathrm{sat}\left(E/\Phi \right)$$
where
(40)$$\mathrm{sat}\left(E/\Phi \right)=\left\{\begin{array}{cc} E/\Phi ,\; & |E|<\Phi ,\\ \mathrm{sgn}(E),\; & |E|\geq \Phi . \end{array}\right.$$
Taking Equations (Equation 33), (Equation 34) and (Equation 39) into Equation (Equation 27) yields
(41)$$\dot{E}+b{\hat{b}}^{-1}k\;\mathrm{sat}\left(E/\Phi \right)=\left(\hat{\alpha }-\alpha \right)f+\left(b-\hat{b}\right)\hat{u}.$$
Based on Equations (Equation 22) and (Equation 23), the following inequalities are obtained:(42)$$\rho_{2}\leq b^{-1}\hat{b}\leq \rho_{1},\;\rho_{1}^{-1}\leq b{\hat{b}}^{-1}\leq \rho_{2}^{-1}$$
where
(43)$$\rho_{1}=\frac{\hat{b}}{\hat{b}-\mathrm{sgn}\left(\hat{b}\right)\bar{b}}>1,\;0<\rho_{2}=\frac{\hat{b}}{\hat{b}+\mathrm{sgn}\left(\hat{b}\right)\bar{b}}<1.$$

Two situations: (1). |E|≥Φ and (2). |E|<Φ are discussed below.

(1). |E|≥Φ

Equations (Equation 38) and (Equation 41) lead to
(44)$$k|E|\geq b^{-1}\hat{b}\left(\hat{\alpha }-\alpha \right)fE+\left(1-b^{-1}\hat{b}\right)\hat{b}\hat{u}E+b^{-1}\hat{b}\left(\eta -\dot{\Phi }\right)\left|E\right|.$$
Because
(45)$$b^{-1}\hat{b}\left(\hat{\alpha }-\alpha \right)fE\leq \rho_{1}\bar{\alpha }|f||E|,$$
(46)$$\left(1-b^{-1}\hat{b}\right)\hat{b}\hat{u}E\leq \left(1-\rho_{2}\right)|\hat{b}\hat{u}||E|,$$
(47)$$b^{-1}\hat{b}\left(\eta -\dot{\Phi }\right)\left|E\right|\leq \left\{\begin{array}{cc} \rho_{1}\left(\eta -\dot{\Phi }\right)\left|E\right|,\; & \eta \geq \dot{\Phi },\\ \rho_{2}\left(\eta -\dot{\Phi }\right)\left|E\right|,\; & \eta <\dot{\Phi }, \end{array}\right.$$
*k* can be set as
(48)$$k=\left\{\begin{array}{cc} \rho_{1}\bar{\alpha }\left|f\right|+\left(1-\rho_{2}\right)\left|\hat{b}\hat{u}\right|+\rho_{1}\left(\eta -\dot{\Phi }\right),\; & \eta \geq \dot{\Phi },\\ \rho_{1}\bar{\alpha }\left|f\right|+\left(1-\rho_{2}\right)\left|\hat{b}\hat{u}\right|+\rho_{2}\left(\eta -\dot{\Phi }\right),\; & \eta <\dot{\Phi }. \end{array}\right.$$

(2). |E|<Φ

Equation (Equation 41) yields a first-order low-pass filter on the compact error *E*:(49)$$\dot{E}+\frac{b{\hat{b}}^{-1}k}{\Phi }E=\left(\hat{\alpha }-\alpha \right)f+\left(b-\hat{b}\right)\hat{u}.$$
The filter’s input (right side of the above equation) can be interpreted as the perturbation due to parameter uncertainties. Because all the parameters are bounded, the compact error *E* will be stabilized in a finite time. The filter’s cut-off frequency is
(50)$$f_{c}=\frac{b{\hat{b}}^{-1}k}{\Phi }\leq \frac{k}{\rho_{2}\Phi }=\lambda_{1},$$
where $\lambda_{1}$ is the frequency limit we set to make the filter work properly. Higher value of cut-off frequency results in lower filter gain, which is beneficial to stabilize *E*, while it also might excite unmodeled high-frequency dynamics. There, the frequency limit $\lambda_{1}$ is set as high as possible within the bandwidth of the plant model. Equation (Equation 50) results in
(51)$$k=\lambda_{1}\rho_{2}\Phi .$$
Equations (Equation 48) and (Equation 51) together are used to tune Φ and *k* online (initial values: Φ(0)=0, k(0)=0), which works as a first-order exponentially stable filter, leading to the global boundedness of Φ and *k*. With Φ and *k* being determined, the input command *u* is obtained according to Equations (Equation 33), (Equation 34) and (Equation 39). The establishment of the sliding mode controller is complete.

### 2.3. Control Strategy

The layers of adaption and robustness presented above make it is possible to design the control system based on a reformed control plant instead of the original control plant. The reformed control plant could be totally different and much more simpler compared to the original control plant, where their differences are compensated by the control system. In this way, the efforts to determine the control plant for a complicated system as precise as possible could be saved, and the design process of the control system could be significantly simplified.

In the classic sliding mode control where no estimator is implemented, the plant’s parameters need to be bounded in known ranges; thus, the control plant should be carefully modeled. The proposed control strategy does not require that, but on the other hand, it results in an issue that the estimation error boundaries $\bar{\alpha }$ and $\bar{b}$ cannot be determined prior to deploying the control system. To address this issue, $\bar{\alpha }$ and $\bar{b}$ can be set as relatively large values or updated online based on the estimations $\hat{\alpha }$ and $\hat{b}$, e.g., by using
(52)$$\bar{\alpha }=0.9|\hat{\alpha }|,\;\bar{b}=0.9\left|\hat{b}\right|.$$
This approach is conservative and can cause a high value of correction gain *k* which would make the controller very aggressive if no boundary layer is adopted. Fortunately, large $\bar{\alpha }$ and $\bar{b}$ also results in large boundary layer to increase the probability that the low-pass filter on *E* works, and the filter’s cut-off frequency $f_{c}$ will not change much according to Equation (Equation 50). Therefore, the control system would still work properly. The proposed control strategy is demonstrated by the following illustrative example.

### 2.4. An Illustrative Example

A forced Van der Pol oscillator is considered in this illustrative example:(53)$$\ddot{x}=-\alpha_{v1}x-\alpha_{v2}\left(x^{2}-1\right)\dot{x}+b_{v}u.$$
The true values of parameters are
(54)$$\alpha_{v1}\left(t\right)=40+4\;\sin\left(\frac{2\pi }{3}t+\frac{\pi }{4}\right),$$
(55)$$\alpha_{v2}\left(t\right)=2+0.2\;\sin\left(\frac{4\pi }{3}t-\frac{\pi }{4}\right),$$
(56)$$b_{v}\left(t\right)=1+0.1\;\sin\left(\frac{\pi }{2}t+\frac{\pi }{6}\right),$$
which are unknown to us. In the following, we will implement control systems based on three different control plants. The first one is the *original plant* given by Equation (Equation 53); the second and third ones are
(57)$$\ddot{x}=-\alpha_{r11}x-\alpha_{r12}\dot{x}+b_{r1}u,$$
and
(58)$$\ddot{x}=-\alpha_{r2}x+b_{r2}u,$$
respectively, which are referred to as *reformed plant 1* and *reformed plant 2*. The reformed plant 1 replaces the nonlinear item $\alpha_{v2}\left(x^{2}-1\right)\dot{x}$ in Equation (Equation 53) with a linear item $\alpha_{r12}\dot{x}$, and the reformed plant 2 takes a further step to remove that nonlinear item.

The control systems adopt same parameters, which are
(59)$$\theta_{0}=200,\;k_{0}=4000,\;P\left(0\right)=\mathrm{diag}\left(k_{0},...,k_{0}\right)$$
for the BLS estimator, and
(60)$$\lambda =\lambda_{1}=100,\;\eta =0.1$$
for the sliding mode controller. The initial estimations of the original plant’s parameters are set as
(61)$${\hat{\alpha }}_{v1}\left(0\right)=40,\;{\hat{\alpha }}_{v2}\left(0\right)=2,\;{\hat{b}}_{v}\left(0\right)=1,$$
which are close to the true values, assuming that we have a basic understanding of the original plant. However, there is no such information for the reformed plants; therefore, the initial estimations of the reformed plants’ parameters are simply set as
(62)$${\hat{\alpha }}_{r11}\left(0\right)={\hat{\alpha }}_{r12}\left(0\right)={\hat{b}}_{r1}\left(0\right)=1,$$
(63)$${\hat{\alpha }}_{r2}\left(0\right)={\hat{b}}_{r2}\left(0\right)=1.$$
The designated displacement trajectory $x_{d}$ for this example is a randomly generated band-limited white noise with cut-off frequency: 3 Hz, power spectral density: $400\;{\mathrm{mm}}^{2}$, sampling frequency: 1024 Hz and time duration: 8 s. The designated velocity trajectory ${\dot{x}}_{d}$ is obtained by taking derivative operation on $x_{d}$.

The displacement and velocity tracking results of the three control plants are shown in Figure 1. All three control plants could result in a good tracking performance without obvious differences among them.

To quantitatively compare three control plants, displacement and velocity relative tracking errors (abbreviated to dis. err. and vel. err.):(64)$$e_{x}\left(t\right)=\frac{x\left(t\right)-x_{d}\left(t\right)}{\max[x_{d}\left(t\right)]}\times 100\%,\;e_{\dot{x}}\left(t\right)=\frac{\dot{x}\left(t\right)-{\dot{x}}_{d}\left(t\right)}{\max[{\dot{x}}_{d}\left(t\right)]}\times 100\%$$
are calculated. The peak dis. and vel. err. of three control plants are listed in Table 1. It can be inferred from Table 1 that the peak errors of reformed plants are kept at a low level (less than 1%) and are close to those of the original plant. Additionally, the reformed plant 2 even achieves better displacement tracking performance than the original plant. Therefore, both reformed plants can be used for the design of control systems.

The true values and estimations of the original plant’s parameters are shown in Figure 2. Although there are oscillations existing in the estimations, the variations of parameters are generally well tracked by the BLS estimator. For the reformed plants, the parameter estimations are shown in Figure 3 and Figure 4, which are varying in large ranges:(65)$${\hat{\alpha }}_{r11}\in \left[-115.79,190.32\right],\;{\hat{\alpha }}_{r12}\in \left[-7.90,15.86\right],\;{\hat{b}}_{r1}\in \left[0.42,1.60\right],$$
and
(66)$${\hat{\alpha }}_{r2}\in \left[-207.30,226.9\right],\;{\hat{b}}_{v2}\in \left[0.18,1.81\right].$$
The model errors of the reformed plants are essentially compensated by the large oscillations of their parameters. It is important to remark that the parameter ranges of the reformed plant 2 is larger than those of the reformed plant 1. The reason behind this observation is that the reformed plant 2 is more simple than the reformed plant 1; thus, higher parameter variations of the reformed plant 2 are required to compensate the plant errors. The numerical example conducted in this section proves that the proposed control strategy is feasible.

## 3. Application to Real-Time Hybride Simulation

Figure 5 is the block diagram of a general RTHS. The entire structure of interest is divided into the numerical substructure modeled on a computer and the physical substructure manufactured for testing. The measured force from the physical substructure is fed into the numerical substructure to calculate the designated motions (e.g., displacement, velocity or acceleration) at the interface/boundary between the numerical and physical substructures. This loop is referred to as the *outer loop*. The control system generates the command signal based on the designated and measured motions it received and sends the command signal to the transfer system (normally the servo-hydraulic actuator), which then applies dynamic loads to the physical substructure to make it move along the designated motions. This loop is referred to as the *inner loop*.

The physical plant in an RTHS is very complicated, because it includes the dynamics of the transfer system, physical substructure and control–structure interaction (CSI) [21,22] between them, with the existence of parametric/non-parametric modeling errors, measurement noises, time delays/lags, etc. Additionally, the RTHS performance is evaluated not by simply checking the tracking performance of the control system in the inner loop, but by comparing the results of the RTHS with those of the *reference model* of the entire structure (reference structure). A small tracking error in the inner loop has a considerable effect to deviate the designated motions in the outer loop, which might eventually significantly deteriorate the RTHS performance. Therefore, the performance requirement of the RTHS control system is much more stringent than that of the normal control system which only has one loop.

A virtual RTHS benchmark problem on the seismic analysis of a linear three-story steel frame [14] has been proposed for the community to enable researchers to evaluate their RTHS control systems. Figure 6 depicts the RTHS reference structure and its partitioning scheme, where a nonlinear MR damper is added into the structure in this paper to further increase the complexity of the physical plant. Note that in the default benchmark problem definition, there is no damper, and the whole structure is linear. In the following sections, we will present the control system design based on the control strategy we proposed and evaluate the performance of the control system.

### 3.1. Reference Structures

The modeling of the reference structure where no damper is installed has been given in the definition paper of the RTHS benchmark problem [14], which is omitted here. The MR damper we add is assumed to be governed by a force–velocity model [23]:(67)$$F_{MR}=k_{d}A_{n}\tan^{-1}\left(n_{0}{\dot{x}}_{MR}\right)+k_{d}C_{n}{\dot{x}}_{MR}$$
where
(68)$$A_{n}=q_{1}+(q_{2}\left|v\right|+q_{3}{\left|v|\right)}^{1/3},$$
(69)$$C_{n}=r_{1}+(r_{2}\left|v\right|+r_{3}{\left|v|\right)}^{1/3},$$
$F_{MR}$ (unit: kN) and ${\dot{x}}_{MR}$ (unit: m/s) are the force and velocity, respectively, *v* (Unit: V) is the input voltage of the damper, $n_{0}$, $q_{1}$, $q_{2}$, $q_{3}$, $r_{1}$, $r_{2}$ and $r_{3}$ are constant model parameters whose values are listed in Table 2 and $k_{d}$ is the scale coefficient we add to adjust the damper force. In this paper, we set $k_{d}=6.5\times 10^{-3}$ to let the damper contribute around 20% of the first floor’s force. Figure 7 plots the force–displacement and force–velocity responses of the MR damper under sinusoidal displacement input: 5sin2.5πt (mm), when the input voltage is 0.0, 0.5 and 1.5 V.

To establish the model of the reference structure, which is installed with the MR damper, the damper force $F_{MR}$ obtained from Equation (Equation 67) is added into the first floor’s force. In the benchmark problem definition, three earthquakes, El Centro 1940, Kobe 2005 and Morgan Hill 1984, and a linear chirp signal (0∼10 Hz, 90 s) are selected as the ground acceleration inputs of the structure, with their amplitudes being scaled down to 40%. Because a fixed specimen is used as the physical substructure in the RTHS, four different partitioning cases are created by changing the floor mass (three floors have identical mass) and damping ratio (three modes have identical damping ratio) of the structure. The four partitioning cases are listed in Table 3. The predictive stability indicator (PSI) can be used to indicate the sensitivity of the partitioning case [24,25]. Based on the PSIs of the four cases, it has been confirmed that Case 4 is the most sensitive case among them and needs to be carefully dealt with in the RTHS [14].

In this paper, we consider three different reference structures, as explained in Table 4, where Intact w/o d is the default reference structure in the benchmark problem definition. Figure 8 compares the third-floor displacement responses of Intact w/o d and Intact w d under the El Centro earthquake for Case 4. It can be seen that the structural response is significantly reduced when the MR damper is installed. Figure 9 compares the third-floor displacement responses of Intact w/ d and Damaged w d under the El Centro earthquake for Case 4. The response of the intact structure is sharply increased compared to that of the damaged structure after the damage occurs.

### 3.2. Physical Plant and Reformed Plant

Figure 10 is the block diagram of the physical plant for Intact w/ d, where $y_{Gc}$ is the command input sending to the transfer system; $F_{m}$ and $x_{m}$ are the measurements with sensor noises of the the physical substructure’s force *F* and displacement *x*; $m_{e}$, $c_{e}$ and $k_{e}$ are the physical substructure’s mass, damping coefficient and stiffness; $a_{1}\beta_{0}$, $a_{2}$, $a_{3}$, $\beta_{1}$ and $\beta_{2}$ are parameters associated with the dynamics of the servo-hydraulic actuator. For Intact w/o d, the block of the MR damper model is disabled at the beginning of the simulation. For Damaged w/ d, the block of the MR damper model is disabled and $c_{e}$ and $k_{e}$ are decreased to 10% at 14 s to simulate the sudden damage.

For Intact w/o d, the physical plant is linear, and the transfer function from $y_{Gc}$ to *x* is given by [14]:(70)$$G_{p}\left(s\right)=\frac{B_{0}}{A_{5}s^{5}+A_{4}s^{4}+A_{3}s^{3}+A_{2}s^{2}+A_{1}s+A_{0}}$$
where
(71)$$\begin{array}{cc} \; & B_{0}=a_{1}\beta_{0},\;A_{0}=k_{e}a_{3}\beta_{2}+a_{1}\beta_{0},\\ \; & A_{1}=k_{e}a_{3}\beta_{1}+\left(k_{e}+c_{e}a_{3}+a_{2}\right)\beta_{2},\\ \; & A_{2}=k_{e}a_{3}+\left(k_{e}+c_{e}a_{3}+a_{2}\right)\beta_{1}+\left(c_{e}+m_{e}a_{3}\right)\beta_{2},\\ \; & A_{3}=k_{e}+c_{e}a_{3}+a_{2}+\left(c_{e}+m_{e}a_{3}\right)\beta_{1}+m_{e}\beta_{2},\\ \; & A_{4}=c_{e}+m_{e}a_{3}+m_{e}\beta_{1},\;A_{5}=m_{e}. \end{array}$$
Equation (Equation 70) is defined as the *physical plant* of Intact w/o d. The nominal values and the associated uncertainties (standard deviations) of the plant’s parameters are listed in Table 5 [14].

In our previous work [17], a reformed plant:(72)$$G_{pr}\left(s\right)=\frac{1}{s^{2}+\alpha_{2}s+\alpha_{1}}\times \frac{d_{1}\left(1+d_{2}s\right)}{1+d_{3}s}.$$
was developed to fit Equation (Equation 70). In the reformed plant, $d_{1}$, $d_{2}$ and $d_{3}$ are constants, while $\alpha_{1}$ and $\alpha_{1}$ are assumed to bounded, i.e.,
(73)$$|\alpha_{1}-{\hat{\alpha }}_{1}|={\bar{\alpha }}_{1},\;\left|\alpha_{2}-{\hat{\alpha }}_{2}\right|={\bar{\alpha }}_{2}$$
to consider uncertainties, where ${\hat{\alpha }}_{1}$ and ${\hat{\alpha }}_{2}$ are nominal values of $\alpha_{1}$ and $\alpha_{1}$, and ${\bar{\alpha }}_{1}$ and ${\bar{\alpha }}_{2}$ are their boundaries. The nominal values and boundaries of these parameters are listed in Table 6.

Figure 11 compares the frequency responses of the physical and reformed plants for Intact w/o d. It can be seen from Figure 11 that the reformed plant is capable to encompass the dynamic behaviors of the physical plant over a wide frequency range. However, it is important to remark that the good fitting shown in Figure 11 is only for Intact w/o d. The physical plants of Intact w/ d and Damage w/ d are highly nonlinear, and they have much different dynamic properties compared to that of Intact w/o d, as shown in Figure 8 and Figure 9. Additionally, they can hardly be expressed by explicit equations for the design of model-based control systems. In this paper, we will not form new reformed plants for physical plants of Intact w/ d and Damage w/ d. Instead, we will develop a unique control system which works for Intact w/o d, Intact w/ d and Damage w/ d.

### 3.3. Control System Design

The adjust command *u* is defined as
(74)$$u=\frac{d_{1}\left(1+d_{2}s\right)}{1+d_{3}s}y_{Gc}.$$

Combining Equations (Equation 72) and (Equation 74) yields the adjust plant:(75)$$\ddot{x}=-\alpha_{1}x-\alpha_{2}\dot{x}+u,$$
which is a second-order linear system. The control system design in this section is based on the adjust plant. The parameters $\alpha_{1}$ and $\alpha_{2}$ are estimated online by the BLS estimator, and the nominal values of them listed in Table 6 are adopted as the initial estimations: ${\hat{\alpha }}_{1}\left(0\right)$ and ${\hat{\alpha }}_{2}\left(t\right)$. To make the control system robust to different reference structures, $\alpha_{1}\left(t\right)$ and $\alpha_{2}\left(t\right)$ are assumed to be varying in relatively large boundaries:(76)$${\bar{\alpha }}_{1}={\hat{\alpha }}_{1}\left(0\right),\;{\bar{\alpha }}_{2}={\hat{\alpha }}_{2}\left(0\right)$$
instead of the boundaries listed in Table 6. Note that it has been verified by the authors that if the boundaries listed in Table 6 are used for $\alpha_{1}\left(t\right)$ and $\alpha_{2}\left(t\right)$, the control system still works well, while larger boundaries are preferred which could achieve a higher control performance.

#### 3.3.1. BLS Estimator

Equation (Equation 75) cannot be directly adopted for the BLS estimator, because only the first state *x* is measurable, and the measurement $x_{m}$ contains noise. To eliminate the states $x={\left[x,\dot{x}\right]}^{T}$ and reject the noise effect, Equation (Equation 74) is multiplied by a second-order low-pass filter:(77)$$\frac{1}{s^{2}+z_{2}s+z_{1}}$$
on both sides. Then, the linear estimation equation y=WΨ of the BLS estimator is obtained:(78)$$\underset{y}{\underset{︸}{x_{m}-\frac{1}{s^{2}+z_{2}s+z_{1}}u}}=\underset{W}{\underset{︸}{\left[-\frac{1}{s^{2}+z_{2}s+z_{1}}x_{m},\;-\frac{s}{s^{2}+z_{2}s+z_{1}}x_{m}\right]}}\times \underset{\Psi }{\underset{︸}{\left[\begin{array}{c} \alpha_{1}-z_{1}\\ \alpha_{2}-z_{2} \end{array}\right]}}$$
where the state *x* is replaced by the measurement $x_{m}$. With the estimation $\hat{\Psi }\left(t\right)$, ${\hat{\alpha }}_{1}\left(t\right)$ and ${\hat{\alpha }}_{2}\left(t\right)$ are calculated by using
(79)$$\left[{\hat{\alpha }}_{1}\left(t\right),{\hat{\alpha }}_{2}\left(t\right)\right]=\hat{\Psi }{\left(t\right)}^{T}+\left[z_{1},z_{2}\right].$$

Adding a low-pass filter in a control loop may introduce an instability problem; thus, the filter should be carefully designed. The low-pass filter’s parameters are selected as $z_{1}=1600\;s^{-2}$ and $z_{2}=10\;s^{-1}$ to put our focus on the main frequency range 0∼10 Hz of the system response, as shown in Figure 12. The BLS estimator’s parameters are set as $\theta_{0}=10$ and $k_{0}=1\times 10^{6}$.

#### 3.3.2. Time-Varying Kalman Filter

The estimations ${\hat{\alpha }}_{1}\left(t\right)$ and ${\hat{\alpha }}_{2}\left(t\right)$ are used to deploy a time-varying Kalman filter to obtain the state estimations $\hat{x}={\left[\hat{x},\dot{\hat{x}}\right]}^{T}$ based on the measurement $x_{m}$. The state-space representation of Equation (Equation 75) is
(80)$$\begin{array}{cc} \; & \dot{x}=A\left(t\right)x+Bu\\ \; & x_{m}=Cx+w \end{array}$$
where
(81)$$A\left(t\right)=\left[\begin{array}{cc} 0 & 1\\ -{\hat{\alpha }}_{1}\left(t\right) & -{\hat{\alpha }}_{2}\left(t\right) \end{array}\right],\;B=\left[\begin{array}{c} 0\\ 1 \end{array}\right],\;C=\left[1\;0\right],$$
and *w* is the white noise with covariance of $2.56\times 10^{-10}\;m^{2}$, which is given in the benchmark problem definition. A Simulink block *Kalman filter* is adopted to calculate the filter gain L(t) online and estimate the states based on the following law
(82)$$\dot{\hat{x}}=A\left(t\right)\hat{x}+Bu+L\left(t\right)\left(x_{m}-C\hat{x}\right).$$
Because A(t) and L(t) are updated with ${\hat{\alpha }}_{1}\left(t\right)$ and ${\hat{\alpha }}_{2}\left(t\right)$, Equation (Equation 82) represents a time-varying Kalman filter.

It is important to remark that the parameter estimations of the plant are obtained first, and then they are used in the time-varying Kalman filter to estimate the plant’s states. This process is achievable because the plant is linear; thus, we could design a low-pass filter as Equation (Equation 76) to make the BLS estimator work solely based on the measurement $x_{m}$. For a complex nonlinear plant, the BLS estimator normally requires full states without noises, which will lead to a contradiction because the BLS estimator should be executed before the full states are estimated by the states filter. This problem deserves more studies, and it is out of scope of this paper.

#### 3.3.3. Sliding Mode Controller

The measured force $F_{m}$ is treated as one of the inputs of the numerical substructure (the other input is the ground acceleration) to calculate the designated motions $x_{d}$, ${\dot{x}}_{d}$ and ${\ddot{x}}_{d}$ of the physical substructure. Note that although the measurement $F_{m}$ also contains sensor noise, the negative effect of noise can be well eliminated because the numerical substructure works like a low-pass filter. The simulation time step is τ=1/4096s; thus, a phase-lead compensator
(83)$$e^{\tau s}\approx \frac{1+2\tau s}{1+\tau s}$$
is implemented to compensate the one-time-step delay in $x_{d}$, ${\dot{x}}_{d}$ and ${\ddot{x}}_{d}$.

The compact error is
(84)$$E=\dot{\hat{x}}-{\dot{x}}_{d}+\lambda \left(\hat{x}-x_{d}\right).$$
Because there is no parameter uncertainty on the input gain of the adjust plant, the control law can be deduced as
(85)$$u=\hat{u}+u_{c},$$
where
(86)$$\hat{u}={\hat{\alpha }}_{1}\hat{x}+{\hat{\alpha }}_{2}\dot{\hat{x}}+{\ddot{x}}_{d}+\lambda ({\dot{x}}_{d}-\dot{\hat{x}}),\;u_{c}=-k\;\mathrm{sat}\left(E/\Phi \right),$$
and adaptation laws of the correction gain *k* and the boundary layer Φ are
(87)$$k=\lambda_{1}\Phi ={\bar{\alpha }}_{1}|\hat{x}|+{\bar{\alpha }}_{2}\left|\dot{\hat{x}}\right|+\eta -\dot{\Phi },\;k\left(0\right)=\Phi \left(0\right)=0.$$
The parameters of the sliding mode controller are set as η=0.1, $\lambda =\lambda_{1}=30\;\mathrm{Hz}=188\;s^{-1}$.

The control system design is complete. The implementation of the control system is shown in Figure 13.

### 3.4. Results and Discussion

In the execution of the virtual RTHS (pure simulation), the physical plant that uses the nominal parametric values listed in Table 5 is defined as the *nominal plant*, and the physical plant that uses uncertain parametric values (randomly generated according to the standard deviations) is defined as the *perturbed plant*. The third-floor displacement responses of Intact w/o d, Intact w/ d and Damaged w/ d, where the El Centro earthquake, partitioning Case 4 and an identical perturbed plant are used, are shown in Figure 14, Figure 15 and Figure 16. It can be seen from these figures that high RTHS tracking performances are achieved. The peak relative displacement tracking errors for the three reference structures are 0.62, 3.60 and 2.67%, respectively. For Intact w/o d, the physical plant is linear and is well represented by the reformed plant; thus, the tracking error is kept at a much lower level compared to Intact w/ d. For Intact w/ d, the tracking error is oscillating irregularly during the entire simulation. The reason is that the BLS estimator and the sliding mode controller are trying to instantaneously compensate the unmodeled nonlinear behaviors of the plant. For Damaged w/ d, the plant is restored to be linear after 14 s due to the failure of the MR damper. The control system takes around two seconds to adapt the sudden plant change caused by the damage and then works efficiently to decrease the tracking error to a low level.

The parameter estimations for the different reference structures are shown in Figure 17. There are no ground accelerations in the first five seconds; thus, the BLS estimator is actually not activated during that time due to the insufficiency of the data. After 5 s, the BLS estimator starts to work and adapt to the parameter variations. The estimated parameters for Intact w/o d are converged to constants because the physical plant is linear. The estimated parameters for Intact w/o d are time varying to compensate for the nonlinear dynamics created by the additional MR damper. For Damaged w/ d, the estimated parameters change dramatically during the two seconds (14∼16 s) after the damage occurs and then converge to constants. The observations from Figure 14 to Figure 17 demonstrate that the unique control system is robust and adaptive to three different reference structures. Furthermore, the sudden change in the plant of Damaged w/ d can be reflected in the parameter estimations, which provides a potential way to monitor the structural condition of the plant during the execution of the RTHS.

Nine evaluation criteria $J_{1}∼J_{9}$ need to be calculated according to the benchmark problem definition. $J_{1}$, $J_{2}$ and $J_{3}$ are used to evaluate the control performance in the inner loop. $J_{1}$ is the tracking time delay, and $J_{2}$ and $J_{3}$ are the normalized root mean square error and peak error for the displacement tracking. $J_{4}∼J_{9}$ are a set of relative errors between the structural responses of the RTHS and those of the reference structures to evaluate the RTHS global tracking performance. The detailed descriptions and expressions of $J_{1}∼J_{9}$ can be found in the definition document [14].

Table 7, Table 8, Table 9 and Table 10 list the evaluation criteria of Damaged w/ d under the El Centro earthquake. For each partitioning case, the nominal plant and five perturbed plants are considered. All the tracking delays ($J_{1}$) remain at zero, and $J_{2}$ and $J_{3}$ are no greater than 0.61%. Therefore, the control system achieves an excellent inner-loop tracking performance. $J_{4}∼J_{9}$ are slightly higher than $J_{2}$ and $J_{3}$, while they are still kept at a low level (1.79∼4.06%), indicating a good RTHS global performance. It can be seen that the evaluation criteria for Case 4 are higher than those for the other partitioning cases, which agrees with the conclusion that Case 4 is the most sensitive partitioning case. The evaluation criteria of Damaged w/ d under the Kobe and Morgan earthquakes and the Chirp excitation for Case 4 are listed in Appendix A. The evaluation criteria of Intact w/o d and Intact w/ d also have low values (less than 5%), which are not listed in this paper for readability.

## 4. Conclusions

The control system design for the RTHS is more challenging compared to a normal control problem because there is an additional outer loop in the RTHS, and the physical control plant might be complicated, nonlinear and normally hard to obtain or cannot be explicitly expressed. In this paper, a control strategy is proposed for the RTHS, where the control system is designed based on a reformed plant which is highly simplified compared to the physical plant. The control system is featured with adaption and robustness, fulfilled by the BLS estimator and the sliding mode controller, respectively. The reformed plant can be linear with low orders and only needs to capture the basic dynamic properties of the physical plant. The unmodeled plant dynamics are compensated online by the BLS estimator and the sliding mode controller during the execution of the RTHS.

An RTHS benchmark problem is studied to demonstrate the effectiveness of the proposed control strategy. Three reference structures are considered in this paper: (1) Intact w/o d, where the structure is intact and no damper is installed, (2) Intact w/ d, where the structure is intact and a nonlinear MR damper is installed, and (3) Damaged w/d, the MR damper is installed, and the structure is damaged at 14 s. These reference structures result in much different physical plants, while a single control system is designed for these reference structures based on a unique reformed plant. The simulation results show that the control system achieves high RTHS global tracking performances for all three reference structures. Therefore, the proposed control strategy for the RTHS is validated. The control strategy could simplify the design process of the control system based on a reformed plant and avoid the effort to determine the physical plant as precisely as possible, making it very efficient and practical for the RTHS.

## Figures and Tables

**Figure 1 sensors-22-06569-f001:**
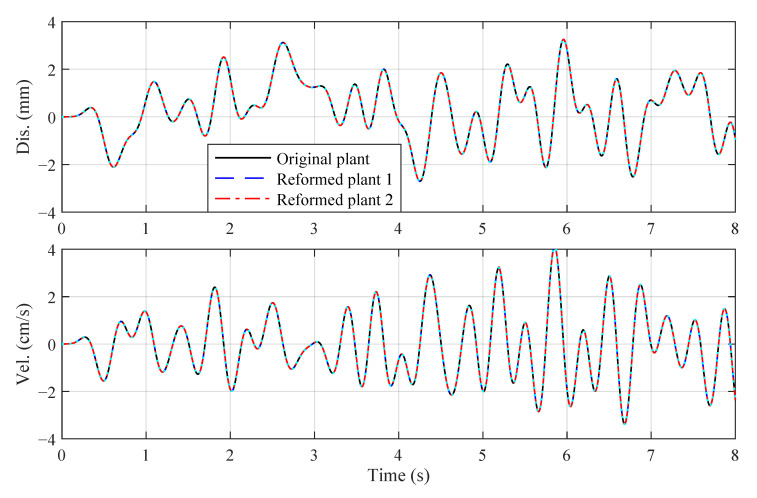
Tracking results for the forced Van der Pol oscillator.

**Figure 2 sensors-22-06569-f002:**
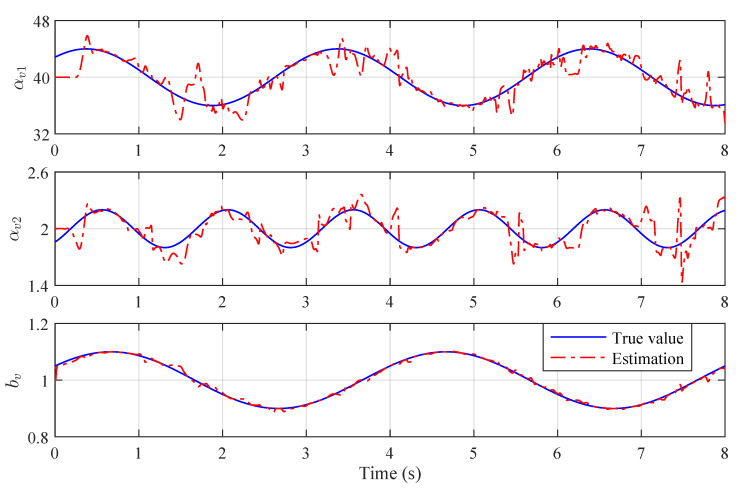
True values and estimations of the original plant’s parameters.

**Figure 3 sensors-22-06569-f003:**
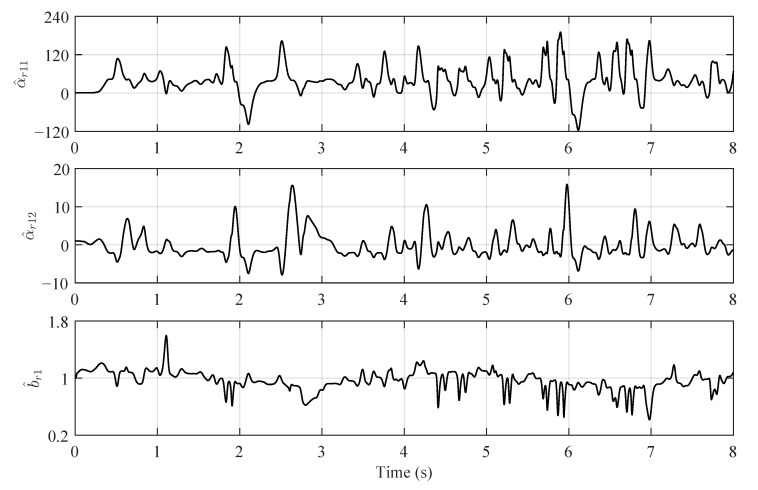
Estimations of the reformed plant 1’s parameters.

**Figure 4 sensors-22-06569-f004:**
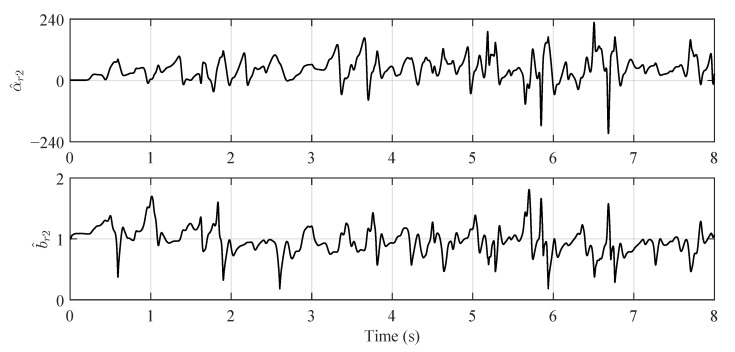
Estimations of the reformed plant 2’s parameters.

**Figure 5 sensors-22-06569-f005:**
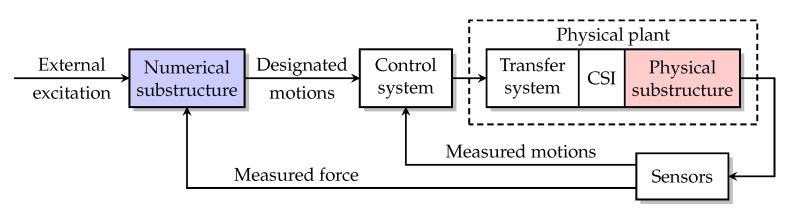
RTHS block diagram.

**Figure 6 sensors-22-06569-f006:**
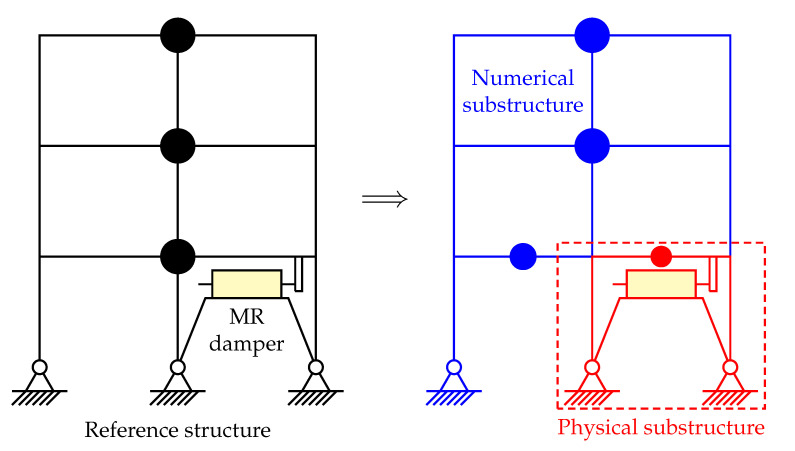
RTHS partitioning scheme.

**Figure 7 sensors-22-06569-f007:**
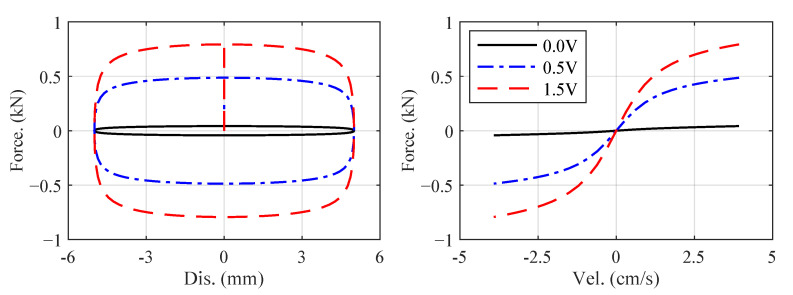
Force–displacement and force–velocity responses of the MR damper under sinusoidal displacement input: 5sin2.5πt (mm).

**Figure 8 sensors-22-06569-f008:**
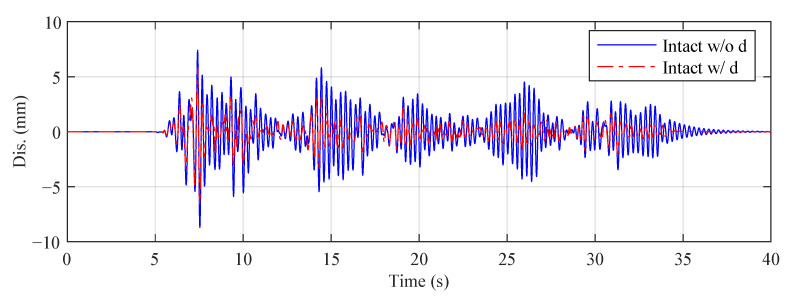
Third-floor displacement responses of Intact w/o d and Intact w/ d (El Centro earthquake, Case 4).

**Figure 9 sensors-22-06569-f009:**
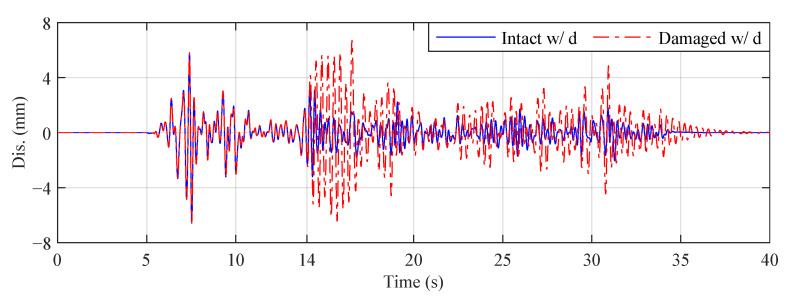
Third-floor displacement responses of Intact w/ d and Damaged w/ d (El Centro earthquake, Case 4).

**Figure 10 sensors-22-06569-f010:**
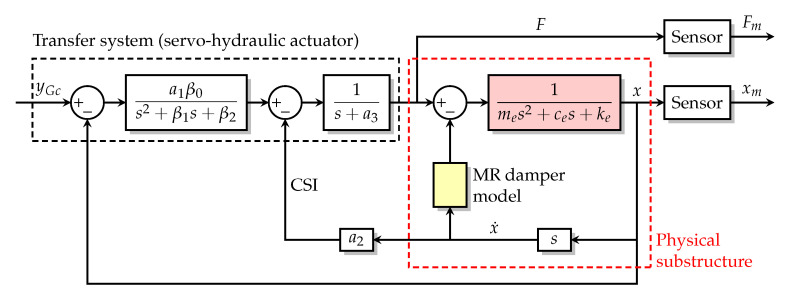
Block diagram of the physical plant for Intact w/ d.

**Figure 11 sensors-22-06569-f011:**
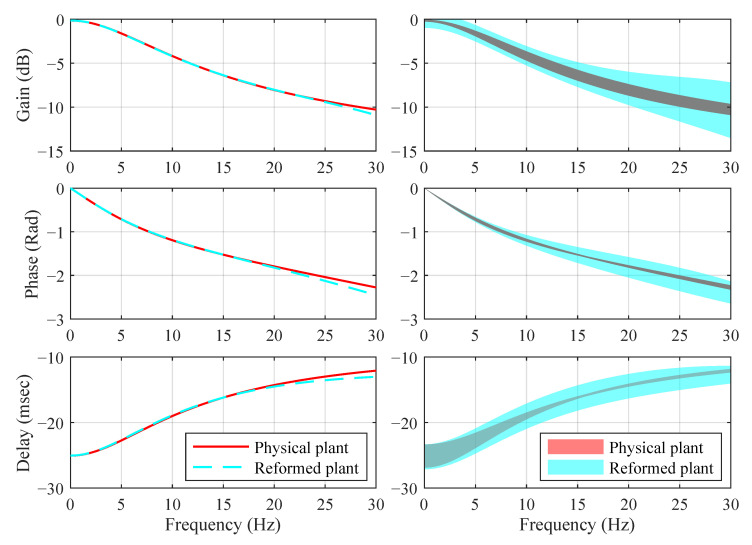
Frequency responses of physical and reformed plants (**left**: nominal models; **right**: models with uncertainties).

**Figure 12 sensors-22-06569-f012:**
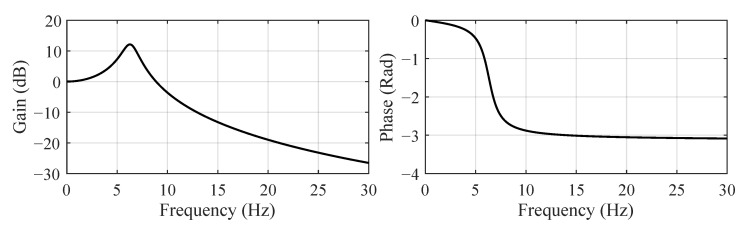
Frequency response of the low-pass filter: $z_{1}=1600\;s^{-2}$, $z_{2}=10\;s^{-1}$.

**Figure 13 sensors-22-06569-f013:**
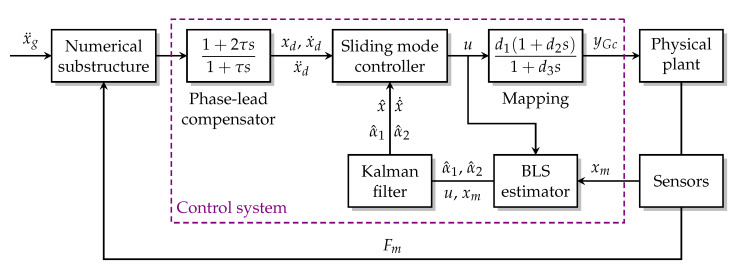
Block diagram of the control system in RTHS.

**Figure 14 sensors-22-06569-f014:**
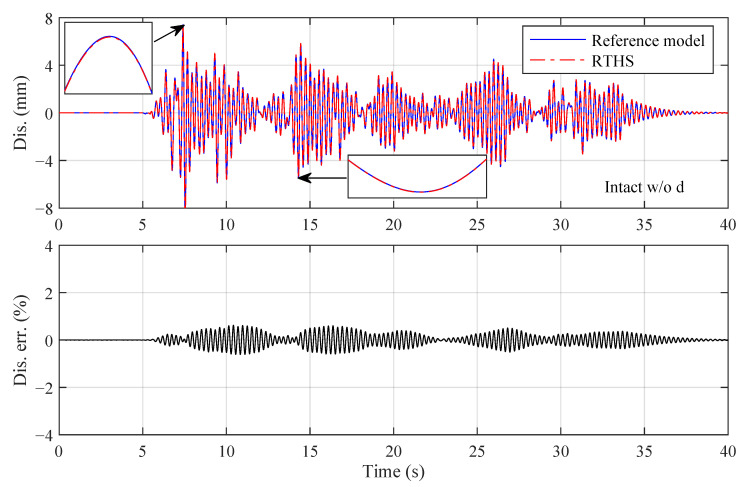
Third-floor displacement responses of Intact w/o d (El Centro earthquake, Case 4, perturbed plant).

**Figure 15 sensors-22-06569-f015:**
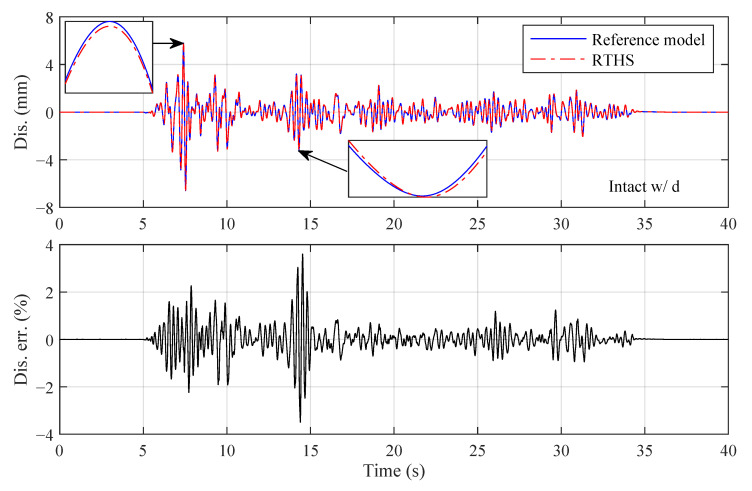
Third-floor displacement responses of Intact w/ d (El Centro earthquake, Case 4, perturbed plant).

**Figure 16 sensors-22-06569-f016:**
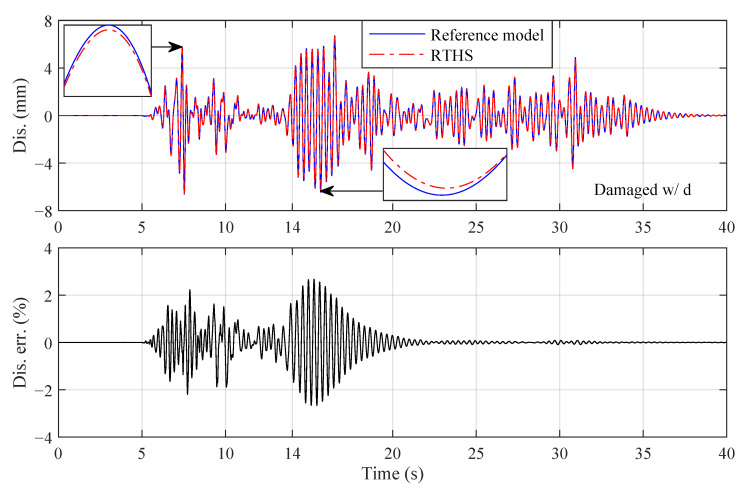
Third-floor displacement responses of Damaged w/ d (El Centro earthquake, Case 4, perturbed plant).

**Figure 17 sensors-22-06569-f017:**
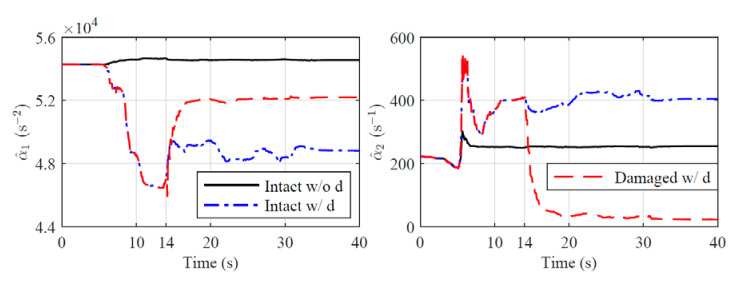
Parameter estimations for Intact w/o d, Intact w/ d and Damaged w/d (El Centro earthquake, Case 4, perturbed plant).

**Table 1 sensors-22-06569-t001:** Peak dis. and vel. err. of three control plants.

	Original Plant	Reformed Plant 1	Reformed Plant 2
Peak dis. err. (%)	0.617	0.673	0.584
Peak vel. err. (%)	0.723	0.881	0.948

**Table 2 sensors-22-06569-t002:** Parametric values of the MR damper model.

$q_{1}$	$q_{2}$	$q_{3}$	$r_{1}$	$r_{2}$	$r_{3}$	$n_{0}$
2.2	$2.16\times 10^{5}$	$7.38\times 10^{4}$	92	$1.58\times 10^{6}$	$2.40\times 10^{5}$	100

**Table 3 sensors-22-06569-t003:** Four RTHS partitioning cases.

Partitioning Case	Floor Mass (kg)	Damping Ratio (%)
Case 1	1000	5
Case 2	1100	4
Case 3	1300	3
Case 4	1000	3

**Table 4 sensors-22-06569-t004:** Three reference structures.

Label	Description
Intact w/o d	The physical substructure is intact, and no damper is installed.
Intact w/ d	The physical substructure is intact, and an MR damper is installed.
Damaged w/ d	An MR damper is installed. The physical substructure is damaged at 14 s.
	When damage occurs, the stiffness and damping of the physical substructure
	drop to 10%, and the MR damper stops working.

**Table 5 sensors-22-06569-t005:** Nominal values and standard deviations of the physical plant’s parameters.

Parameter	Nominal Value	Standard Deviation	Unit
$a_{1}\beta_{0}$	$2.13\times 10^{13}$	—	$\mathrm{kg}/s^{5}$
$a_{2}$	$4.23\times 10^{6}$	—	$\mathrm{kg}/s^{2}$
$a_{3}$	3.3	1.3	$s^{-1}$
$\beta_{1}$	425	3.3	$s^{-1}$
$\beta_{2}$	$1\times 10^{5}$	$3.31\times 10^{3}$	$s^{-2}$
$m_{e}$	29.1	—	kg
$c_{e}$	114.6	—	kg/s
$k_{e}$	$1.19\times 10^{6}$	$5\times 10^{4}$	$\mathrm{kg}/s^{2}$

**Table 6 sensors-22-06569-t006:** Nominal values and boundaries of the reformed plant’s parameters.

Parameter	Nominal Value	Boundary	Unit
$\alpha_{1}$	54,290	5429	$s^{-2}$
$\alpha_{2}$	221.64	79.79	$s^{-1}$
$d_{1}$	53,354	—	$s^{-2}$
$d_{2}$	$1.06\times 10^{-4}$	—	s
$d_{3}$	$2.11\times 10^{-2}$	—	s

**Table 7 sensors-22-06569-t007:** Evaluation criteria (Damaged w/ d, El Centro earthquake, Case 1).

Control	$J_{1}$	$J_{2}$	$J_{3}$	$J_{4}$	$J_{5}$	$J_{6}$	$J_{7}$	$J_{8}$	$J_{9}$
Plant	(msec)	(%)	(%)	(%)	(%)	(%)	(%)	(%)	(%)
Nominal	0	0.45	0.62	2.77	2.67	2.87	2.86	2.74	2.62
Pert. 1	0	0.45	0.62	2.93	2.86	3.03	3.02	2.92	2.79
Pert. 2	0	0.44	0.62	2.88	2.94	2.97	2.95	2.98	2.84
Pert. 3	0	0.45	0.62	2.71	2.78	2.79	2.78	2.82	2.69
Pert. 4	0	0.45	0.62	2.54	2.47	2.63	2.62	2.54	2.43
Pert. 5	0	0.45	0.62	2.91	2.79	3.00	2.99	2.86	2.73

**Table 8 sensors-22-06569-t008:** Evaluation criteria (Damaged w/ d, El Centro earthquake, Case 2).

Control	$J_{1}$	$J_{2}$	$J_{3}$	$J_{4}$	$J_{5}$	$J_{6}$	$J_{7}$	$J_{8}$	$J_{9}$
Plant	(msec)	(%)	(%)	(%)	(%)	(%)	(%)	(%)	(%)
Nominal	0	0.43	0.61	2.95	3.21	3.01	3.00	3.19	3.07
Pert. 1	0	0.43	0.61	2.95	3.27	2.97	2.96	3.22	3.10
Pert. 2	0	0.44	0.61	2.61	2.83	2.66	2.66	2.82	2.72
Pert. 3	0	0.44	0.61	2.58	2.82	2.63	2.62	2.80	2.70
Pert. 4	0	0.44	0.61	2.83	3.03	2.90	2.89	3.02	2.91
Pert. 5	0	0.44	0.61	2.51	2.50	2.61	2.61	2.57	2.48

**Table 9 sensors-22-06569-t009:** Evaluation criteria (Damaged w/ d, El Centro earthquake, Case 3).

Control	$J_{1}$	$J_{2}$	$J_{3}$	$J_{4}$	$J_{5}$	$J_{6}$	$J_{7}$	$J_{8}$	$J_{9}$
Plant	(msec)	(%)	(%)	(%)	(%)	(%)	(%)	(%)	(%)
Nominal	0	0.40	0.57	2.38	2.18	2.33	2.34	1.97	1.89
Pert. 1	0	0.40	0.57	2.44	2.25	2.40	2.40	2.04	1.92
Pert. 2	0	0.40	0.57	2.34	2.14	2.29	2.29	1.93	1.85
Pert. 3	0	0.40	0.56	2.24	1.96	2.20	2.20	1.79	1.81
Pert. 4	0	0.41	0.57	2.28	2.07	2.21	2.21	1.86	1.79
Pert. 5	0	0.40	0.57	2.43	2.25	2.39	2.40	2.04	1.92

**Table 10 sensors-22-06569-t010:** Evaluation criteria (Damaged w/ d, El Centro earthquake, Case 4).

Control	$J_{1}$	$J_{2}$	$J_{3}$	$J_{4}$	$J_{5}$	$J_{6}$	$J_{7}$	$J_{8}$	$J_{9}$
Plant	(msec)	(%)	(%)	(%)	(%)	(%)	(%)	(%)	(%)
Nominal	0	0.46	0.56	3.62	3.74	3.75	3.74	3.94	3.86
Pert. 1	0	0.45	0.55	3.78	3.86	3.91	3.90	4.06	4.04
Pert. 2	0	0.45	0.55	3.61	3.74	3.73	3.71	3.91	3.89
Pert. 3	0	0.46	0.59	3.40	3.56	3.54	3.53	3.68	3.53
Pert. 4	0	0.46	0.56	3.42	3.61	3.54	3.53	3.79	3.70
Pert. 5	0	0.46	0.57	3.58	3.71	3.71	3.70	3.91	3.80

## Data Availability

The data presented in this study are available on request from the corresponding author. The data are not publicly available due to the privacy restrictions of the ongoing research.

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
