# Peer review of "An Adaptive and Robust Control Strategy for Real-Time Hybrid Simulation"

_sensors, 2022, doi:10.3390/s22176569_

Round 1

Reviewer 1 Report

This paper presents an RTHS control strategy based on adaptive and robust control theories. The approach is formulated with two layers. An illustrative example, the Van der Pol oscillator, is shown to highlight details of the BLS estimator. Finally, the main application is presented in section 3 with results and discussion. The paper is well written, easy to follow, and the figures are of high quality. Before publication, I recommend the authors respond to the following questions and comments. 

  1. What do authors want to say with "normal control"? Please, be more precise. 
  2. I suggest being more specific when referring to the RTHS problem in the abstract and introduction. In the current version, the reader only discovers the type of RTHS when it reaches section 3. 
  3. Equation 76: what if the authors had followed the ranges in table 6? What would happen? 

Reviewer 2 Report

This study proposes a control strategy for real-time hybrid simulation, in particular for a nonlinear structure-control system. In this control strategy, a bounded-gain forgetting least-squares estimator is integrated into a sliding mode controller. An illustrative example is provided to show the implementing procedure of the proposed strategy, while this control strategy is also applied to a benchmark problem for RTHS. As seen in the results, the proposed control strategy achieves high tracking performance.

This manuscript is well prepared and very detailed. Before this manuscript is recommended for publication in this journal, the following comments should be addressed.

1.      Adding a low-pass filter in a control loop may introduce an instability problem. The authors may justify the use of low-pass filters in the Conclusions.

2.      More insights should be provided in the Abstract, e.g., both bounded-gain forgetting least-squares estimator and sliding mode controller.

3.      In Eq. (2), the superscript of the state variables should be defined.

4.      Eq. (8) is not a formal equation.

5.      “higher weights on the small range of recent data points would enlarge the noise and disturbance effects” may be incorrect. The reasons should be the modeling errors and uncertainties.

6.      In Line 121, the power spectral density should have a unit.

7.      It would be better if those variables in Table 2 had units.

8.      The errors in the vertical axis of Figure 14 follow the same definition in Eq. (64)? If yes, then the errors (Err.) should be changed into dis. err.; otherwise, the definition for the calculation of errors should be provided. This comment can also be applied to Figures 15-16.

Reviewer 3 Report

This paper proposes a novel RTHS control strategy by combining theories of adaptive control and robust control, where a reformed plant which is highly simplified compared to the physical plant can be used to design the control system without compromising control performance. This control strategy is validated through the investigation of a RTHS benchmark problem.

This study provides novel and interesting results. The paper is very well organized. The addition of the following recent publications can help to enrich the References part: 

-"A European Association for the Control of Structures joint perspective. Recent studies in civil structural control across Europe". Structural Control and Health Monitoring. 2014 Dec;21(12):1414-36. Doi:10.1002/stc.1652

-"Evaluation of time delay margin for added damping of SDOF systems in real-time dynamic hybrid testing (RTDHT) under seismic excitation". In Proceedings of the Fifteenth World Conference on Earthquake Engineering, Lisbon, Portugal, 2012.

-"Exploring the effects of tuned mass dampers on the seismic performance of structures with nonlinear base isolation systems". Earthquakes and Structures. 2017;12(3):285-96. Doi:10.12989/eas.2017.12.3.285
